# Development and characterization of amino donor-acceptor Stenhouse adducts

Cesar A. Reyes[1], Hye Joon Lee[1], Connie Karanovic[1] & Elias Picazo [1] ✉

Donor-acceptor Stenhouse adducts (DASAs) are molecular photoswitches spurring wide interest because of their dynamic photophysical properties, complex photoswitching mechanism, and diverse applications. Despite breakthroughs in modularity for the donor, acceptor, and triene compartments, the backbone heteroatom remains static due to synthetic challenges. We provide a predictive tool and sought-after strategy to vary the heteroatom, introduce amino DASA photoswitches, and analyze backbone heteroatom effects on photophysical properties. Amino DASA synthesis is enabled by aza-Piancatelli rearrangements on pyrrole substrates, imparting an aromaticity-breaking rearrangement that capitalizes on nitrogen's additional bonding orbital and the inductive properties of sulfonyl groups. Amino DASA structure is confirmed by single crystal X-ray diffraction, the photochromic properties are characterized, and the photoswitch isomerization is investigated. Overall, the discovered pyrrole rearrangement enables the study of the DASA backbone heteroatom compartment and furthers our insight into the structure-property relationship of this complex photoswitch.

Since their discovery as photoresponsive materials in 2014[1], donor–acceptor Stenhouse adducts (DASAs) have emerged as a prominent class of photoswitches with the capacity to advance smart materials, phototherapy, and logic-gated systems, among other applications (Fig. 1a)[2–5]. Studies have elucidated the structure-property relationships and multi-state switching mechanism that lead to absorbance in the visible to near-IR region, negative photochromism, and significant volume and polarity changes upon switching[6]. Adaptable DASA physical properties arise from their assembly (Fig. 1b)[7–9], which introduces modularity in molecular structure (Fig. 1c). However, since furan substrates have been the only heterocycles to undergo aza-Piancatelli rearrangements, the backbone heteroatom has remained limited to hydroxy substitution.

Second-generation DASAs evaluated the donor compartment and expanded to include aromatic amine donors such as indoline (Fig. 1c)[10,11]. The extended conjugation provided by second-generation donors decreases charge separation within the open isomer, resulting in faster switching kinetics and broader solvent compatibility[12,13]. Third-generation DASAs explored the acceptor compartment, and it

was determined that stronger pull character leads to a bathochromic shift in absorbance and better control over dark equilibria, allowing for over 95% open-to-closed isomerization upon irradiation[14]. More recently, substitutions on the triene compartment were studied. Aryl and bromide substitutions resulted in bathochromic shifts that produced DASAs with absorbance in the near-IR region[15]. Further, alkyl substitutions were shown to create steric interactions that encourage closed-to-open thermal reversion[16]. The impact of each molecular compartment on DASA physical properties and potential applications inspires a community-wide interest in discovering the contributions from the backbone heteroatom compartment.

Interested in the switching mechanics, Feringa and coworkers synthesized a non-hydroxy triene derivative via Zincke salt formation followed by secondary amine addition and Knoevenagel condensation to investigate the backbone heteroatom effects[17]. The non-hydroxy triene was found to isomerize more slowly than the DASA by approximately one order of magnitude. Further, the non-hydroxy triene underwent photoisomerization at either C2−C3 or C3−C4, as opposed to exclusive rotation about C2−C3 for the parent DASA

[1]Department of Chemistry, Loker Hydrocarbon Research Institute, University of Southern California, 837 Bloom Walk, Los Angeles, CA, USA.
✉e-mail: epicazo@usc.edu

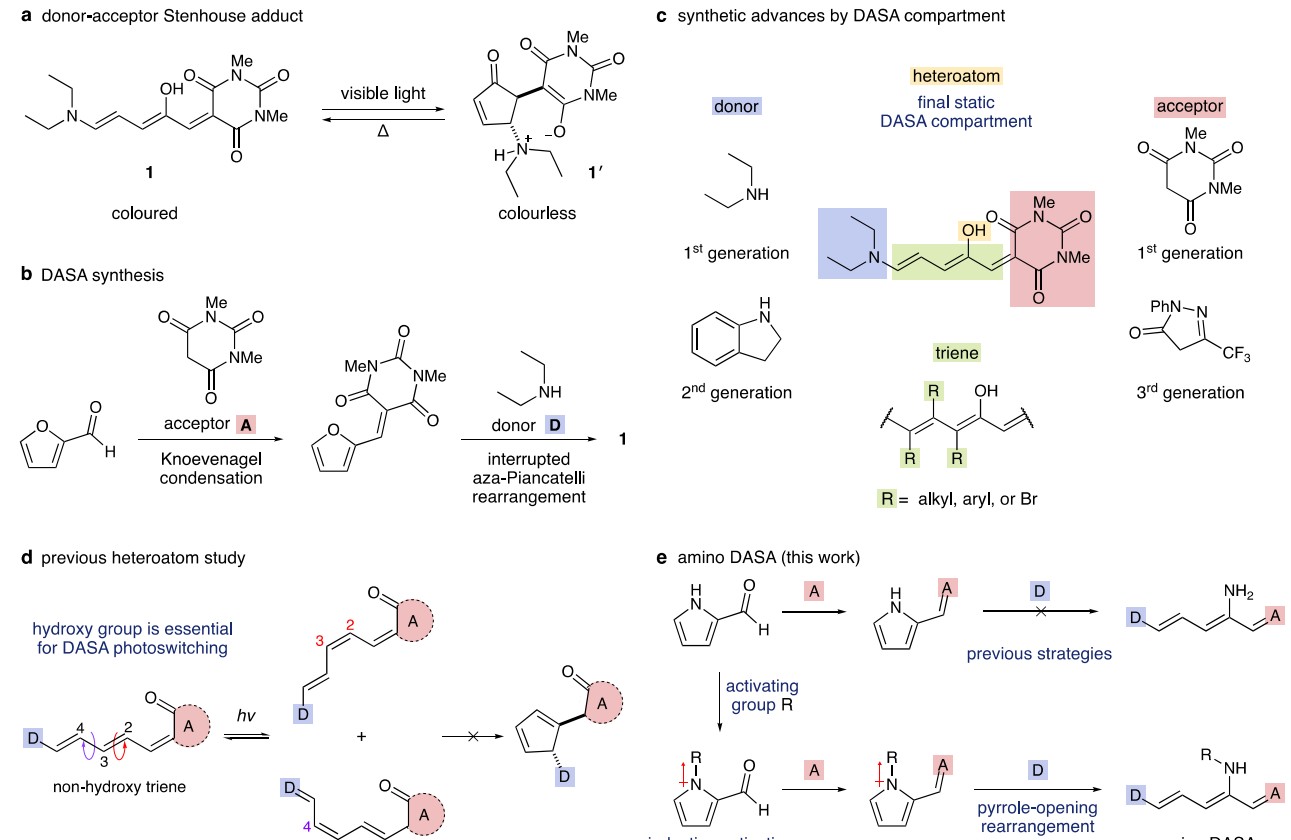

**Fig. 1 | Donor–acceptor Stenhouse adducts: overview and aim of study.**
**a** Isomerization of the donor–acceptor Stenhouse adduct **1** and **1'**. **b** Knoevenagel condensation of 2-furaldehyde with a carbon acid acceptor followed by furan ring opening with an amine donor to form DASA compounds. **c** Synthetic advances of DASA structure per compartment. **d** Vital backbone hydroxy group for DASA composition highlighted by failure to undergo 4π-electrocyclization with non-hydroxy triene. **e** Development of amino DASAs by leveraging nitrogen's additional bonding orbital to promote a pyrrole rearrangement.

(Fig. 1d). Computed potential energy curves for the cyclization step point towards an energetically forbidden pathway for the non-hydroxy triene. Together these results highlight a key hydrogen bond between the backbone hydroxy group and the acceptor carbonyl, as well as stabilizing electronic effects on the triene backbone provided by the heteroatom. The hydroxy group's vital role in DASA switching mechanics urges the targeted synthesis of DASAs with varying backbone heteroatomic properties. As a result, we identified amino DASAs as leading candidates for their ability to enable backbone heteroatom studies.

Pioneering furan rearrangements dating back to as early as 1850 on furfurals by Stenhouse[18], Honda[19], and Lewis and Mulquiney[20], furylcarbinols by Piancatelli[21] and Read de Alaniz[22,23], and activated furans by Šafář[24] and Read de Alaniz[1], have contributed to the synthesis of Stenhouse salt derivatives, a multitude of prostanoic acid natural products[25], and DASA molecular photoswitches. Despite their impact, each rearrangement variant has remained limited to furan cores, constraining products to oxygen-containing molecules. Specific to DASAs, only a hydroxy group has been achievable for the backbone heteroatom compartment due to the synthetic dependence on activated furans. Lalevée and Dumur demonstrated that parent pyrrole and thiophene derivatives that would introduce other heteroatom functionality fail to undergo ring opening[26]. This observation is consistent with heterocycle aromatic stability where thiophene, pyrrole, and furan carry stabilization energies of 120 kJ/mol, 89 kJ/mol, and 66 kJ/mol, respectively[27,28]. Parallels between the Piancatelli rearrangement and DASA synthesis start from nucleophilic amine addition onto the heteroatom-adjacent carbon of the heterocycle (Supplementary Fig. 1). Proton transfer is followed by ring-opening, which

results in the formation of the triene backbone. This triene is the isolable DASA product in its open form. Upon photoisomerization, the thermally stable, open isomer **1** transforms into a cyclopentenone moiety **1'** (Fig. 1a) through a conrotatory 4π-electrocyclization, likening the aza-Piancatelli process. Hence, open-form DASA synthesis relies on an interrupted aza-Piancatelli rearrangement, and the backbone hydroxy group is introduced from the use of a furan starting material.

We hypothesized that nitrogen's additional bonding orbital could be used to inductively activate pyrrole heterocycles to develop an aromaticity-breaking rearrangement. Realization of such a transformation would enable the reinvention of an amino equivalent for all hydroxy permutations since 1850, including the more recent DASA syntheses. Herein, we describe the strategy and success of such a rearrangement and its application for the synthesis of amino DASA photoswitches that enable backbone heteroatom compartment studies (Fig. 1e).

## Results and discussion
### Development of amino DASAs
Polarization of the carbon-oxygen bond that is to be cleaved following nucleophile addition is pivotal to furan ring-opening rearrangements. As such, Lewis acids are commonly employed as catalysts to facilitate Piancatelli rearrangements[29]. Despite the extensive use of lanthanide (III) catalysts for various Piancatelli rearrangements, it was discovered that Dy(OTf)$_3$ only increases DASA production by 10% yield[7]. This observation, along with dysprosium's high degree of oxaphilicity[30], led us to consider the alternative approach of using highly polar protic solvents. Polar protic solvents capable of forming H-bonding networks

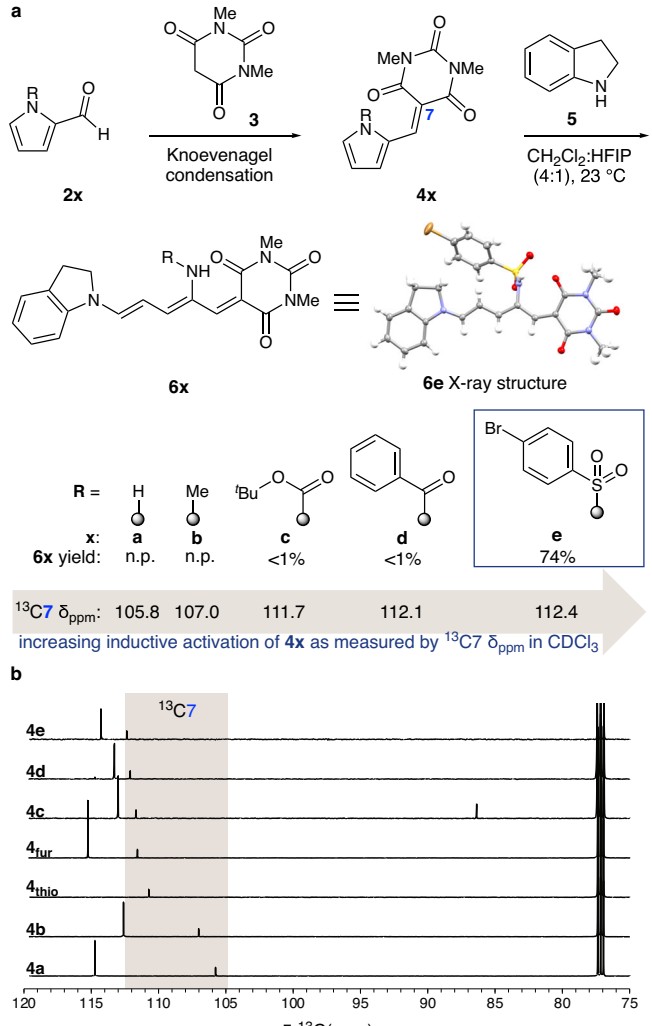

**Fig. 2 | Amino DASA synthesis and a $^{13}$C NMR predictive tool. a** Amino DASA synthetic route, pyrrole-based aza-Piancatelli rearrangement optimization by inductive activation, and characterization of **6e** by single crystal X-ray diffraction. Yields shown are representative of the aza-Piancatelli rearrangement. For additional experimental detail, see the Supplementary Information. **b** A synthetic predictive tool for rearrangement viability based on $^{13}$C NMR shifts of activated heterocycles.

have been used to promote Piancatelli rearrangements[31], and importantly, hexafluoroisopropanol (HFIP) has been shown to provide a remarkable yield increase in DASA synthesis[32]. The application of HFIP as a cosolvent expanded the library of donor amines capable of forming DASA molecules[10,11]. Unfortunately, much like the (aza)-Piancatelli rearrangement, furan has been the only heterocycle reported to undergo such reactivity.

We initiated our efforts to develop a pyrrole-based rearrangement with the goal to access amino photoswitches by synthesizing a range of pyrrole-2-carboxaldehyde substrates **2x** (Fig. 2a). We leveraged nitrogen's additional bonding orbital to tune the degree of electron deficiency and pyrrole's susceptibility to aromaticity-breaking rearrangements. These substrates were then condensed with carbon acid acceptor 1,3-dimethylbarbituric acid (**3**) to afford activated pyrroles **4x**. Lastly, the activated pyrroles were treated with indoline (**5**) donor in a 4:1 mixture of CH$_2$Cl$_2$:HFIP. Full experimental details and reaction optimization studies are available in the Supplementary Information. As observed by Lalevée and Dumur, pyrrole **4a** fails to yield amino DASA product **6a**. Methyl pyrrole **4b** also fails to yield amino DASA product **6b**, ruling out a problematic unprotected pyrrole

and supporting our electronic hypothesis. Boc pyrrole **4c** and benzoyl pyrrole **4d** are reactive but unstable, yielding <1% desired amino DASAs **6c** and **6d**, respectively. Excitingly, we discovered that sulfonyl pyrrole **4e** is reactive and stable under reaction conditions, yielding the desired photochromic molecule in 74% yield. Amino DASA **6e** was recrystallized and the structure of the open, linear form was confirmed by single crystal X-ray diffraction (Deposition Number 2312996, Cambridge Crystallographic Data Centre).

Given the community's interest in modulating the final static component on DASAs and efforts to open thiophene and pyrrole heterocycles for this purpose, we devised a synthetic predictive tool based on $^{13}$C NMR (Fig. 2b). When characterizing the series of activated heterocycles **4x** in CDCl$_3$, we discerned that the $^{13}$C NMR shift of C7, which is in conjugation with the pyrrole nitrogen, is representative of the heterocycle's electronic profile. Specifically, unreactive pyrroles **4a** and **4b** have C7 $^{13}$C NMR shifts of 105.8 and 107.0 ppm, respectively, while **4c** and **4d**, which afford trace levels of desired DASA product **6x**, have C7 $^{13}$C NMR shifts of 111.7 and 112.1 ppm, respectively. For reference, thiophene and furan equivalents, **4$_{thio}$** and **4$_{fur}$**, of pyrrole **4x** have C7 $^{13}$C NMR shifts of 110.7 and 111.6 ppm, respectively. Notably, **4$_{fur}$** successfully undergoes the ring-opening rearrangement while **4$_{thio}$** fails. Given these data, we postulate that a stable activated heterocycle with a C7 $^{13}$C NMR shift of 111.0 ppm or greater is positioned to undergo ring opening. Boc and benzoyl pyrroles **4c** and **4d** do not reliably produce amino DASAs because they are unstable under the reaction conditions. Specifically, reaction of **4c** yields a complex mixture of products while reaction of **4d** produces **4a** via debenzoylation in 99% yield. Consistent with these postulations, sulfonyl pyrrole **4e**, which undergoes ring opening to yield desired amino DASA in 74% yield, has a C7 $^{13}$C NMR shift of 112.4 ppm. Hence, sulfonyl groups are ideal for the synthesis of amino DASA photoswitches because they offer sufficient electron withdrawing aptitude and remain intact under the reaction conditions.

## Characterization of amino DASAs

The developed pyrrole rearrangement granted access to amino DASAs which were then used to understand DASA backbone heteroatom effects. To this end, we synthesized an amino DASA of each generation along with their hydroxy DASA equivalents (Fig. 3a–c). First-generation amino DASA **7** containing dialkyl amine donor isoindoline was synthesized in 43% yield and hydroxy DASA **9** was prepared in 46% yield. It should be noted that other dialkyl amines were explored but found to be incompatible with HFIP cosolvent, consistent with previous hydroxy DASA reports[32]. No desired reactivity is observed in the absence of HFIP, pointing towards a strong dependence on the H-bond network to sufficiently polarize the C−N bond. Second-generation amino DASA **6e** and hydroxy counterpart **10**, which contain aromatic amine donor indoline, were produced in 74% and 70% yield, respectively. Lastly, third-generation DASAs that contain the more strongly withdrawing trifluoromethyl pyrazolone acceptor group were synthesized in 87% and 68% yield for amino DASA **8** and hydroxy DASA **11**, respectively. Thus, our strategy provides comparable yields of the respective rearranged products and a facile means to vary the final compartment on DASA photoswitches.

With representative amino and hydroxy DASAs from each synthetic generation in hand, we set out to characterize their respective physical properties. Initially, we observed that amino DASAs display the same general absorbance trends as parent hydroxy DASAs (Fig. 3d). Specifically, when comparing **7** to **6e**, a $\lambda_{max}$ shift from 531 nm to 578 nm is observed. The bathochromic shift resulting from an aryl amine donor is similarly observed for hydroxy DASAs **9** and **10**, with $\lambda_{max}$ = 573 nm and 615 nm, respectively. Exchanging the acceptor from 1,3-dimethylbarbituric acid to trifluoromethyl pyrazolone results in a bathochromic shift from $\lambda_{max}$ = 578 nm to 608 nm for amino DASAs **6e** and **8** and $\lambda_{max}$ = 615 nm to 646 nm for hydroxy DASAs **10** and **11**,

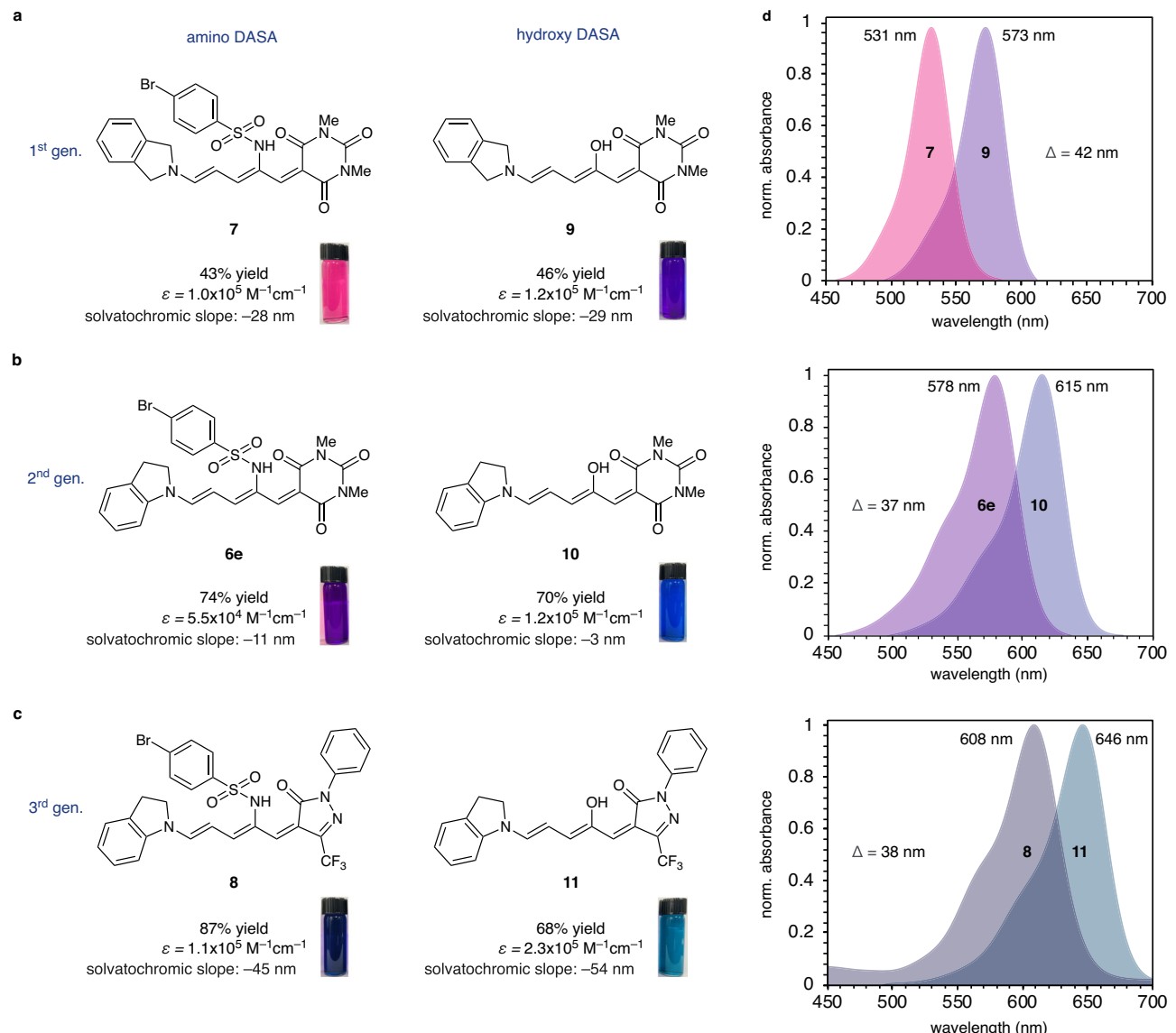

**Fig. 3 | Photophysical characterization of 1st, 2nd, and 3rd generation DASAs.** **a** 1st generation amino DASA **7** and hydroxy DASA **9**, **b** 2nd generation amino DASA **6e** and hydroxy DASA **10**, and **c** 3rd generation amino DASA **8** and hydroxy DASA **11**. Yields are isolated yields. $\varepsilon$ = molar absorption coefficient in $CH_2Cl_2$.

Solvatochromic analyses were performed with normalized Dimroth–Reichardt $E_T^N$ parameters[35]. For additional experimental detail, see Supplementary Fig. 8 in the Supplementary Information. **d** UV-visible absorbance spectra are at 10 μM in $CH_2Cl_2$.

respectively. The bathochromic shifts from aromatic amine donors and strong carbon acid acceptors are due to more diffuse electron delocalization that decreases molecular HOMO-LUMO energetic gaps. Despite similar absorbance shifts of sulfonamide and hydroxy-substituted DASAs with varying donors and acceptors, it was noted that **7** → **6e** resulted in a 12% greater $\lambda_{max}$ shift than **9** → **10** and a 3% lesser $\lambda_{max}$ shift when comparing **6e** → **8** to **10** → **11**. The significant absorbance shift between **7** and **6e** signals that the sulfonamide group may be partaking in noncovalent interactions with the donor or imparting electronic effects that amplify changes in the HOMO-LUMO energetic gap.

When comparing amino DASAs to their hydroxy counterparts, it was observed that sulfonamide containing DASAs express an average hypsochromic absorbance shift of 39 nm. Specifically, **9** to **7** has a hypsochromic shift of 42 nm, while both **10** to **6e** and **11** to **8** have a shift of 37 nm. We suspect that the hypsochromic shift and the larger HOMO-LUMO energetic gap is a direct result of stronger electron induction from the sulfonamide. The withdrawing effects of para-

bromobenzene sulfonamide can be directly contrasted to that of a hydroxyl group through Hammett value analyses. Using 4-substituted benzoic acid derivatives, we calculated[33] substituent constants of $\sigma = 0.13$ for 4-bromobenzene sulfonamide and $\sigma = -0.37$ for hydroxyl. Similarly, 3-substituted benzoic acid derivatives, which remove resonance contributions from the analysis, yield $\sigma = 0.26$ and $0.12$ for 4-bromo sulfonamide and hydroxyl, respectively. These results are in contrast with electronic contributions from other DASA compartments. Weak donors and strong acceptors both lower the HOMO-LUMO energetic gap (vide supra), and electron withdrawing triene substituents also result in a bathochromic shift[18]. It can be postulated that steric interactions of the large sulfonamide may perturb orbital overlap but the X-ray crystal structure of **6e** shows no evidence of steric induced conformational perturbation. Therefore, it seems the backbone heteroatom electronically affects HOMO and LUMO energies distinctly from the other DASA compartments.

In addition to a hypsochromic shift, we found that amino DASAs consistently have a lower molar absorption coefficient than hydroxy

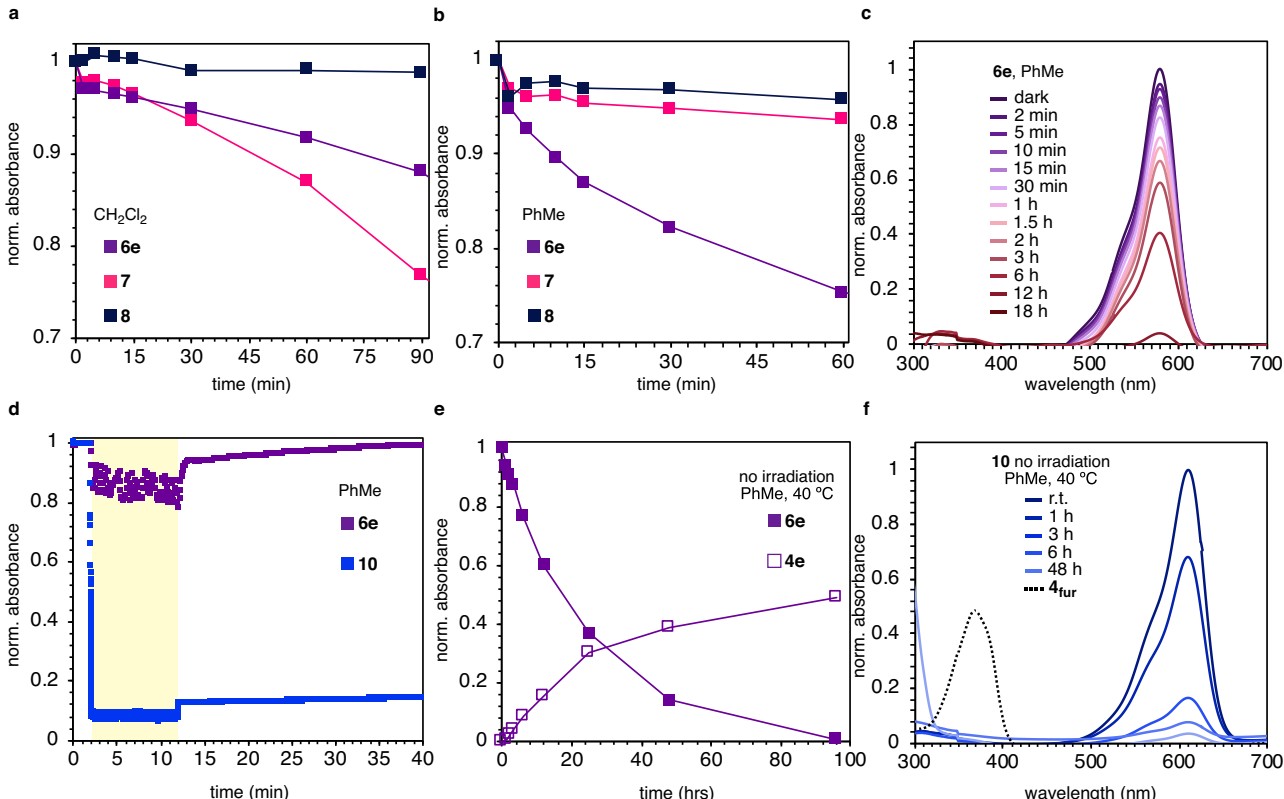

**Fig. 4 | Amino DASA photoisomerization and thermal reversion.** Photoswitching comparison of 1st, 2nd, and 3rd generation amino DASAs in **a** CH$_2$Cl$_2$ and **b** PhMe measured at their corresponding $\lambda_{max}$. **c** Absorbance decrease of amino DASA **6e** and absorbance increase at 381 nm in PhMe upon irradiation. **d** Photoisomerization and thermal reversion of amino DASA **6e** and hydroxy DASA **10** in PhMe. **e** Evaluation of amino DASA **6e** thermal stability. **f** Evaluation of hydroxy DASA **10** thermal stability.

DASAs. While **9** only has a slightly higher coefficient than **7** with $\varepsilon = 1.2 \times 10^5 \, M^{-1} \, cm^{-1}$ compared to $1.0 \times 10^5 \, M^{-1} \, cm^{-1}$, a stronger effect was observed with DASAs containing an aromatic amine donor. Specifically, **6e** has a molar absorption coefficient of $5.5 \times 10^4 \, M^{-1} \, cm^{-1}$ while **10** has one of $1.2 \times 10^5 \, M^{-1} \, cm^{-1}$. A similar two-fold increase was observed for **8** and **11** with $\varepsilon = 1.1 \times 10^5 \, M^{-1} \, cm^{-1}$ and $2.3 \times 10^5 \, M^{-1} \, cm^{-1}$, respectively. The general decrease in absorbance for DASAs containing a sulfonamide substituent highlights the essential role of the backbone heteroatom group in a DASA's photoabsorbent character.

Recent studies shed light on the importance of charge separation in the open DASA isomer and its effect on kinetics and thermal stability[13,34]. The zwitterionic character of amino DASAs and their hydroxy counterparts was measured by solvatochromic analyses using the normalized Dimroth–Reichardt $E_T^N$ parameters (Supplementary Figs. 22–28)[35]. First-generation DASAs show that the backbone heteroatom is seemingly innocent in charge separation, providing solvatochromic slopes of $-28 \, nm/E_T^N$ for **7** and $-29 \, nm/E_T^N$ for **9**. However, such is not the case for second and third-generation DASA photoswitches. Analysis of second-generation DASAs illustrates that amino DASA **6e** has a greater degree of zwitterionic character, yielding a solvatochromic slope of $-11 \, nm/E_T^N$, while hydroxy counterpart **10** is more neutral with a solvatochromic slope of $-3 \, nm/E_T^N$. Contrary, third-generation hydroxy DASA **11** exhibits a higher degree of zwitterionic character, providing a solvatochromic slope of $-54 \, nm/E_T^N$ compared to $-45 \, nm/E_T^N$ for **8**. Although no clear trend is established, it is apparent that the sulfonamide moiety can amplify or attenuate charge separation. This also hints at potential noncovalent interactions between the heteroatom substituent and the donor or acceptor.

After characterizing the photochromic properties of amino DASAs, we investigated the switching capabilities as measured by relative UV-vis absorbance (Fig. 4). Absorbance measurements show that amino DASAs **6e** and **7** respond to irradiation with broadband

visible light in dichloromethane while only **6e** shows absorbance decrease in toluene (Fig. 4a, b). Interestingly, all three generations of amino DASAs have significantly slower conversion rates than their hydroxy counterparts (Supplementary Figs. 29–39). In methanol, amino DASAs undergo absorbance decrease but do not recover absorbance upon ceasing irradiation; this is consistent with hydroxy DASA switching behaviors, with notable exceptions[36,37]. Furthermore, neither **6e** nor **7** display absorbance recovery in any solvent when taken to complete absorbance depletion. When irradiating amino DASAs an absorbance increase in the 300–400 nm range is observed, which correlates with the production of activated pyrrole **4e** ($\lambda_{max} = 381 \, nm$); this effect is shown in Fig. 4c for DASA **6e** in toluene.

To avoid prolonged irradiation, UV-vis absorbance with in situ broadband irradiation across a 10-min interval was measured (Fig. 4d and Supplementary Figs. 37–39). Though hydroxy DASAs project switching characteristics respective to their donor and acceptor compositions, most studied amino DASAs fail to switch under these conditions. Only amino DASA **6e** displays productive reversible photoswitching with a 14% forward photoisomerization upon ten minutes of irradiation in toluene and full recovery after discontinuing irradiation (Fig. 4d). These results are consistent with faster and reversible photoswitching occurring when the backbone heteroatom is more electron donating, further underscoring the importance of the backbone heteroatom in DASA photoswitching behavior.

In contrast to Feringa's non-hydroxy triene[17], amino DASAs displayed complete absorbance decay after extended periods of irradiation. However, given the lack of reversion coupled with an absorbance increase in the 300–400 nm range, the thermal stability of amino DASAs was in question. Upon incubating 2nd generation DASAs in toluene at 40 °C with no irradiation, a significant absorbance decrease is observed. This is not the case for 1st and 3rd generation

DASAs (Supplementary Fig. 18). Upon complete loss of absorbance at $\lambda_{max} = 578$ nm for second-generation amino DASA **6e**, absorbance at 381 nm corresponding to activated pyrrole **4e** increased to 49% conversion (Fig. 4e). In direct contrast, second-generation hydroxy DASA **10** displayed absorbance decrease without observed activated furan **4** _fur_ formation (Fig. 4f).

In effort to observe triene isomerization and cyclization intermediates, we performed a variable temperature NMR in situ irradiation experiment utilizing a narrow band 590 nm LED in deuterated chloroform. Narrow light near $\lambda_{max}$ absorbance excludes potential detrimental effects from broadband exposure. Additionally, amino DASA **6e** displays full absorbance decrease upon irradiation in chlorinated solvents as observed in our UV-vis studies using broadband light (see Supplementary Fig. 29a). However, insignificant changes are observed after 72 h of irradiation with amino DASA **6e** remaining intact (Supplementary Fig. 31), and no isomers are observed when cooling to −15 °C. Therefore, DASAs bearing a more withdrawing sulfonamide group encounter an unproductive C3–C4 cis→trans isomerization in chlorinated solvents at concentrations suitable for NMR acquisition, and prolonged irradiation with broadband light leads to eventual decomposition (Fig. 4a and Supplementary Figs. 29–33). Amino DASA **6e** is insufficiently soluble for variable temperature NMR in situ irradiation experiments in toluene or methanol. Modifications of the backbone heteroatom in combination with additional tuning of other DASA compartments is subject to further investigation.

In conclusion, we report a pyrrole-based aza-Piancatelli rearrangement, a $^{13}$C NMR predictive tool for rearrangement viability, and their application to develop amino donor–acceptor Stenhouse adducts. Confirmed by single crystal X-ray diffraction, amino DASAs enable tunability of the backbone heteroatom, allowing a direct comparison of sulfonamide- and hydroxy-substituted DASAs from each synthetic generation. In comparison to their hydroxy counterpart, amino DASAs produce potential noncovalent interactions in addition to varying electronic contributions that result in a hypsochromic absorbance shift and inefficient photoswitching. It was discovered that substituting the hydroxy group for a sulfonamide moiety results in a decreased molar absorption coefficient and thermal stability, emphasizing the heteroatom's role in efficiently stabilizing the photoswitch. In combination with previous hydroxy DASA studies, we look forward to the future development of well-behaved amino DASAs. With distinct physical properties and an additional bonding orbital, we expect amino DASAs to open opportunities for computational studies and applications to better understand these complex photoswitches.

## Methods
### General considerations
Unless stated otherwise, reactions were conducted in flame-dried glassware using anhydrous solvents (freshly distilled or passed through activated alumina columns). All commercially obtained reagents were used as received unless otherwise specified. Furfural (98%), indoline (98%), and phenylhydrazine (97%) were obtained from Acros Organics B.V.B.A. Pyrrole-2-carboxaldehyde (99%) and *N,N*-dimethylbarbituric acid (99%) were obtained from Beantown Chemical Corporation. Thiophene-2-carboxaldehyde (>98%) and sodium hydride (60% dispersion in oil) were obtained from Tokyo Chemical Industry. Triethylamine (99%) and 4-dimethylaminopyridine (99%) were obtained from Thermo Fisher Scientific. Di-*tert*-butyl dicarbonate (99%), pyridine (99%), and ethyl 4,4,4-trifluoroacetoacetate (99%) were obtained from Sigma-Aldrich. Acetic acid (glacial) and acetic anhydride (99%) were obtained from Ward's Science. 4-bromobenzenesulfonyl chloride (98%) was obtained from Apollo Scientific. 1-methyl-2-pyrrolecarboxaldehyde (98%) was obtained from Lancaster Synthesis Inc. Isoindoline hydrochloride (97%) was obtained

from Ambeed. Hexafluoro-2-propanol (99%) was obtained from Chem-Impex International. Indium (III) bromide (99%) was obtained from STREM Chemicals. Chloroform-*d*, methylene chloride-*d*$_2$ (99.8%), and dimethyl sulfoxide-*d*$_6$ (99.9%) were obtained from Cambridge Isotope Laboratories. Furfural, indoline, and hexafluoro-2-propanol were freshly distilled prior to use. Isoindoline was prepared from isoindoline hyrdrochloride by extracting isoindoline with an alkaline solution (sodium hydroxide). Reaction temperatures were controlled using IKA Plates (RCT digital) and the built-in temperature modulators. Thin layer chromatography (TLC) was conducted with EMD gel 60 F254 pre-coated plates (0.25 mm) and visualized using a combination of UV light, potassium permanganate, phosphomolybdic acid, and p-anisaldehyde staining. Silicycle Silica flash P60 (particle size 0.040−0.063 mm) was used for flash column chromatography. $^1$H NMR spectra were recorded on a Mercury (400 MHz), or Varian spectrometers (500, 600 MHz) and are reported relative to deuterated solvent signals. Data for $^1$H NMR spectra are reported as follows: chemical shift (δ ppm), multiplicity, coupling constant (Hz) and integration. $^{13}$C NMR spectra were recorded on Mercury (100 MHz), or Varian spectrometers (125 MHz, 150 MHz) and are reported relative to deuterated solvent signals. IR data were collected on a Mettler Toledo ReactIR 702 L equipped with a TE MCT detector, an AgX 6 mm × 1.5 m Fiber probe interface, and a DiComp diamond probe tip. All IR data are reported in terms of frequency absorption (cm$^{-1}$). Melting points were recorded on a VWR melting point apparatus, and high resolution mass (HRMS) spectra were obtained on an Agilent 6545Q-TOF LC/MS. UV-visible spectral data was collected on an Agilent Cary 5000 UV-Vis-NIR Spectrophotometer with a UV quartz 10 mm pathlength cuvette (3.5 mL). Crystallographic data was collected on a Rigaku XtaLAB Synergy-S diffractometer.

Irradiation experiments were performed with a Dolan-Jenner Fiber-Lite Model 190, using an EKZ halogen bulb (10.8 V, 30 Watt, 3100 K) via a Dolan-Jenner BGT1826 fiber optic gooseneck on high output for broadband visible light. The convection-cooled temperature control was confirmed with a control experiment replicating DASA irradiation. Over the course of 12 h of irradiation of toluene solution, the temperature changed from 26.2 °C to 26.5 °C without fluctuation.

NMR in situ irradiation experiment was set up following the protocol from Feldmeier et al. (ref. 13). The fiber-optic cable (M28L05; Ø400 μm, 0.39 NA, SMA-SMA Fiber Patch Cable, 5 meters), 590 nm Fiber-coupled LED (M590F3; 4.6 mW, FWHM = 18 nm), 490 nm Fiber-coupled LED (M490F4; 2.8 mW, FWHM = 22 nm), LED Driver (1200 mA), and power supply (KPS201) were purchased from Thor Labs. NMR tubes were purchased from Wilmad-LabGlass (SP Science-ware): WGS-5BL, Coaxial Insert for 5 mm NMR Sample Tube and 535-PP-7, 5 mm Thin Wall Precision NMR Sample Tube 7″ L, 600 MHz. Spectra were measured with a Varian spectrometer (600 MHz).

### Representative procedure for the synthesis of sulfonyl pyrroles
To a flame dried 100 mL round-bottom flask, a magnetic stir bar and NaH (316 mg, 7.89 mmol; 60% wt. in paraffin oil) were added. The flask was flushed with N$_2$, and after adding freshly distilled THF (50 mL 0.1 M) and stirring, it was cooled to 0 °C. Pyrrole-2-carboxaldehyde (500 mg, 5.26 mmol) was then dissolved in a minimal amount of THF (3 mL) and added dropwise. The mixture was then allowed to stir at 0 °C. After stirring for 1 h, 4-bromobenzenesulfonyl chloride (2.00 g, 7.89 mmol) was added, and the mixture stirred at room temperature for 1 h. Then saturated ammonium chloride (50 mL) was added and the aqueous phase was extracted with Et$_2$O (3 × 25 mL). The combined organic phase was washed with brine (50 mL), dried over anhydrous magnesium sulfate, filtered, and concentrated under reduced pressure. The crude oil was purified by flash chromatography (9.6:0.3:0.1 → 9.4:0.5:0.1 → 8.9:1.0:0.1 hexanes:EtOAc:NEt$_3$) to provide pyrrole **2e** (1.51 g, 91% yield) as a white powder.

## Representative procedure for the synthesis of activated sulfonyl pyrroles

In a 10 mL round-bottom flask charged with a magnetic stir bar, pyrrole **2e** (390 mg, 1.241 mmol) and *N,N*-dimethylbarbituric acid (213 mg, 1.365 mmol) were dissolved in 1:1 EtOH:$CH_2Cl_2$ (5 mL) and stirred at room temperature. After 16 h, the yellow precipitate was filtered under vacuum, washed with hexanes, and allowed to dry under high vacuum to provide activated pyrrole **4e** (560 mg, 99% yield) as a yellow solid.

## Representative procedure for the synthesis of amino DASAs

In a 4 mL dram vial charged with a magnetic stir bar, activated pyrrole **4e** (300 mg, 0.663 mmol) was dissolved in 4:1 $CH_2Cl_2$:HFIP (1.1 mL). Indoline (150 μL, 1.326 mmol) was subsequently added and the mixture was stirred at room temperature. An immediate color change to purple was observed. After 1 h the mixture was concentrated under reduced pressure, and the crude residue was recrystallized using $CH_2Cl_2$ as solvent and hexanes as antisolvent to provide amino DASA **6e** (280 mg, 74% yield) as a purple solid. Crystals of **6e** suitable for X-ray diffraction analysis were obtained by recrystallization via vapor diffusion of $Et_2O$ into a solution of **6e** in $CHCl_3$.

## Representative procedure for determination of absorption coefficient and solvatochromic analysis

UV-visible absorption measurements of **6e** were taken at 10, 5, 1, and 0.5 μM in $CH_2Cl_2$ using an Agilent Cary 5000 UV-Vis-NIR Spectrophotometer with a UV quartz 10 mm pathlength cuvette (3.5 mL). Absorbance $\lambda_{max}$ was plotted against concentration (M). The slope of the linear relationship was extrapolated as the molar absorption coefficient for **6e** ($\varepsilon = 5.5 \times 10^4 \, M^{-1} \, cm^{-1}$). Solvatochromic analysis was performed by obtaining the UV-visible absorption measurements of **6e** in PhMe, $Et_2O$, THF, EtOAc, $CHCl_3$, $CH_2Cl_2$, acetone, DMSO, MeCN, and MeOH. The $\lambda_{max}$ of absorbance in each solvent was plotted against the respective polarity value ($E_T^N$) using the Dimroth–Reichardt parameters.

## Data availability

The authors declare that the data supporting the findings of this study are available within the paper and its supplementary information files. Crystal structure data that supports the structure of amino DASA **6e** have been deposited in the Cambridge Crystallographic Data Centre and can be accessed using Deposition Number 2312996 via www.ccdc.cam.ac.uk/data_request/cif. All data may be obtained from the corresponding author upon request.

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

## Acknowledgements
The authors are grateful to the Loker Hydrocarbon Research Institute and the University of Southern California for support. We are also grateful to NIH-NIGMS R00GM140070 for financial support. C.R. thanks the Loker Hydrocarbon Research Institute for the Harold E. Moulton Fellowship. C.K. thanks the NSF-REU (CHE-1757942) program for financial support. We thank S. Guillen and Dr. E. Mcclure for providing HRMS training and X-ray diffraction training, respectively. Instrumentation in the USC Chemistry Instrument Facility was acquired with USC Research and Innovation Instrumentation Award Program, the NSF (DBI-0821671, CHE-0840366, CHE-2018740), and NIH (S10 RR25432) support.

## Author contributions
E.P. conceived of and directed the investigations. C.R., H.L., and C.K. designed the experiments and optimized the reaction methodology. E.P. and C.R. prepared the manuscript with revisions provided by the other authors.

## Competing interests
The Authors declare the following competing interests: provisional patent filed. Patent applicant: University of Southern California; inventors E.P., C.R., and H.L.; US Patent Application No.: 63/567,010 for "Photoresponsive Compounds". Specifically, this patent claims the use of our synthetic route for amino DASAs, amino DASAs themselves, and the predictive tool based on carbon NMR for DASA preparation. C.K. declares no competing interests and all other Authors declare no further competing interests.
