## [Peer Review File · Nature Communications]

Development and characterization of amino donor-acceptor
Stenhouse adductsREVIEWER COMMENTS

Reviewer #1 (Remarks to the Author):

Picazo and co-workers described their efforts toward the development and characterization of amino donor-acceptor Stenhouse adducts as a new subclass of DASA photoswitches. Significantly, these novel amino DASAs are designed and prepared by unprecedented pyrrole-based interrupted aza-Piancatelli rearrangements by judicious introduction of N-sulfonyl protected group to facilitate the key heterocycle ring-opening step. The obtained amino DASA structure was unambiguously confirmed by single crystal X-ray diffraction. The photochromic properties and the photoswitch isomerization were also systematically characterized and investigated. In comparison to classical hydroxyl-substituted DASAs, amino DASAs produce potential noncovalent interactions in addition to varying electronic contributions that result in a hypsochromic absorbance shift and significantly slower photoisomerization and thermal reversion. Meanwhile, substituting the hydroxyl group for a sulfonamide moiety results in a decrease of the photochrome's molar absorption coefficient and thermal stability, emphasizing the heteroatom's role in efficiently stabilizing the photoswitch. These distinct physical properties and an additional bonding orbital of newly developed amino DASAs might open new opportunities for their practical applications. This paper is also well written in a concise way including both manuscript and SI. This reviewer recommends publication in Nature Communications.

Reviewer #2 (Remarks to the Author):

In this work Picazo and co-workers report, the first use of substituted pyrrole adducts to access a sub-class of donor-acceptor Stenhouse adducts (DASAs) where the hydroxyl group is replaced with a sulfonamide. This provides a novel class of DASA based photoswitches with distinct photoswitching properties to the hydroxyl based system. Perhaps more interesting is that it also provides the unique ability to probe the role of hydroxyl group and if this is fully understood these compounds could shift DASA based photoswitches in new directions. The authors do a nice job detailing the key structural features for pyrrole-unit and fully characterizing the sulfonamide based DASA. However, the photoswitching properties of the new DASA adducts is not as well developed. Given this is central to the paper and future use of these materials, more detail regarding the photoswitching properties of the new derivatives should be provided prior to publication. Overall, this work is novel, will be of broad interest to the community and I support publication if the authors can expand the photoswitching studies.

Comments:

1. Currently, only the switching properties for 6e is provided. The authors should include the switching properties for 7 and 8, at least in the SI.

2. It is interesting that the sulfonamide adducts require such long periods of irradiation to access the closed isomer. Presumably, this is due to the higher energy for the 4π electrocyclization and proton transfer. The authors note that at the onset of the switching experiment, the λ_{max} at 567 nm is higher than the corresponding shoulder at 539 nm, but after 30 minutes of irradiation the two peaks are observed at equal normalized absorbance. The author should really explore this property in more detail. The broadband white light irradiation might be detrimental to the photoswitching properties of these derivatives. It might be more efficient if narrow band visible light is used, such as 530 nm. Potentially the longer wavelength light is driving the B/B' intermediate back to A and slowing the overall switching efficiency. Monitoring and plotting the kinetics of the 567 and 532 nm independently would also be helpful.

3. How efficient is the Z to E isomerization?

4. It is not clear if the photoswitch recovers after the long reaction times. The authors note that 6e reaches a photostationary state of 16% after 10 minutes of irradiation and then recovers to 96% open in the dark after 30 minutes. The authors should also report the recovery for DCM, PhMe and MeOH after complete conversion to the closed form (ie. Figure 4b). Does it recover?

5. The authors should show the entire UV-Vis trace from 300 nm to 700 nm for all spectra (at least in the SI). This would prove that the absorbance going down is not due to the formation of the pyrrole 4e. Similar to what is observed in Figure S9.

6. It is not clear what is being shown in Figure 4g. In general, the text and Figures are too small and hard to see. Is the decrease at 40 °C due to the formation of the pyrrole 4e for all solvents or is it due to the formation of closed isomer? Only PhMe is shown in the SI.

7. The authors state, "curiously, 6e underwent gradual loss of absorbance (not photoabsorbance) even in the absence of irradiation, and a similar absorbance decrease was observed for 10, indicating a general thermal equilibrium shift for second generation DASAs." This is not a shift in equilibrium, it is conversion of the DASA adduct to the starting material (pyrrole or activated furan). This should be clarified.

8. Given that heat promotes the conversion to starting material, the authors need to more carefully study how prevalent this reaction is under their photoirradiation conditions. What temperature is being reached upon irradiation with white light under the reaction conditions? Presumably, higher than 40 °C. If so, is starting material being produced? Is this solvent dependent?

9. The NMR for 4a should be expanding in the SI and peak assignments given. It is hard to really see the peaks to clearly see the closed isomer. Is the pyrrole starting observed?

10. In the introduction the authors should expand (briefly) the connection between using a furylcarbinol (aza-Piancatelli and Piancatelli; Read de Alaniz and Piancatelli), furfural (Stenhouse adducts; Stenhouse, Honda, Lewis & Mulquiney) and activated furan (DASA; Safar and Read de Alaniz). This will help place in context the wide-use of furan as the core building block and the unique nature of this work.

Reviewer #3 (Remarks to the Author):

The paper by Reyes et al. reports the synthesis and study of a new type of donor-acceptor Stenhouse adducts (DASA). DASA photoswitches have received considerable attention due to their interesting properties. However, generalization and use of these photoswitches has been typically hampered by the difficult control of the properties and the challenging synthesis of significantly diverse compounds. This paper presents the synthesis of a different type of DASA. The revised synthetic route allowed the preparation of currently unknown amino analogues of the DASA basic core. The modular building of these molecules has allowed the modification of most parts of the structure, leaving untouched the hydroxy group coming from the furan moiety. In this paper, a clever modification and tuning of the synthesis has allowed the preparation of the new amino derivatives. The formal aspects of the manuscript are adequate; well-written text, clear and informative figures, and a good selection of references to put in context the research. The methodology used is also correct and the experiments are well-planned and described in detail.

Unfortunately, despite the efforts shown in this paper, the results are very disappointing. Comparison between hydroxy and amino equivalent leads to poor performance of the new compounds for almost every property: hypsochromic absorbance, slower photoconversion and back-reaction. Thus, beyond the modification in the synthetic route allowing for the preparation of new derivatives, results in this paper are mainly negative from the point of view of the application or use of these DASAs.

Overall, the paper is relevant and should be published, but I do not think that the results shown match the requirements to be published in Nature Communications.

Revisions to NCOMMS-24-12012-T, Development and characterization of amino donor-acceptor Stenhouse adducts

Reviewer #1

Summary:

Picazo and co-workers described their efforts toward the development and characterization of amino donor-acceptor Stenhouse adducts as a new subclass of DASA photoswitches. Significantly, these novel amino DASAs are designed and prepared by unprecedented pyrrole-based interrupted aza-Piancatelli rearrangements by judicious introduction of N-sulfonyl protected group to facilitate the key heterocycle ring-opening step. The obtained amino DASA structure was unambiguously confirmed by single crystal X-ray diffraction. The photochromic properties and the photoswitch isomerization were also systematically characterized and investigated. In comparison to classical hydroxyl-substituted DASAs, amino DASAs produce potential noncovalent interactions in addition to varying electronic contributions that result in a hypsochromic absorbance shift and significantly slower photoisomerization and thermal reversion. Meanwhile, substituting the hydroxyl group for a sulfonamide moiety results in a decrease of the photochrome's molar absorption coefficient and thermal stability, emphasizing the heteroatom's role in efficiently stabilizing the photoswitch. These distinct physical properties and an additional bonding orbital of newly developed amino DASAs might open new opportunities for their practical applications. This paper is also well written in a concise way including both manuscript and SI. This reviewer recommends publication in Nature Communications.

Response: We thank the Reviewer #1 for noting the significant chemical and material contributions. The characterization of these materials was only possible because of the discovery and development of the first pyrrole-based interrupted aza-Piancatelli rearrangement, which has been limited to furans since 1850. Photophysical and switching properties between amino and hydroxy DASAs are significantly different. Lastly, we begin to characterize the backbone heteroatom effect and contributions. Overall, this study opens the field for continued synthetic, mechanistic, computational, and application studies.

Reviewer #2

Summary:

In this work Picazo and co-workers report, the first use of substituted pyrrole adducts to access a sub-class of donor-acceptor Stenhouse adducts (DASAs) where the hydroxyl group is replaced with a sulfonamide. This provides a novel class of DASA based photoswitches with distinct photoswitching properties to the hydroxyl based system. Perhaps more interesting is that it also provides the unique ability to probe the role of hydroxyl group and if this is fully understood these compounds could shift DASA based photoswitches in new directions. The authors do a nice job detailing the key structural features for pyrrole-unit and fully characterizing the sulfonamide based DASA. However, the photoswitching properties of the new DASA adducts is not as well developed. Given this is central to the paper and future use of these materials, more detail regarding the photoswitching properties of the new derivatives should be provided prior to publication. Overall, this work is novel, will be of broad interest to the community and I support publication if the authors can expand the photoswitching studies:

Response: We thank the Reviewer #2 for pointing out the synthetic and material impact of our contributions. Given that the scope of this report is to develop a new pyrrole rearrangement to enable initial backbone heteroatom effect characterization, we agree that studies to further understand the photoswitching properties were required. As such, we performed extensive additional experimentation to address the comments below. We thank Reviewer #2 for advancing our understanding of amino DASAs.

Comments:

1. Currently, only the switching properties for **6e** is provided. The authors should include the switching properties for **7** and **8**, at least in the SI.

Response: We thank the Reviewer #2 for their interest in the switching profiles of all amino DASAs reported. As a result, we conducted switching experiments for amino DASAs **7**, **6e**, and **8** (and their hydroxy DASA counterparts) in dichloromethane, toluene, and methanol. We included the results in the updated version of Figure 4 and in the Supplementary Information as Supplementary Figs. 9–17. In combination with comment 5, we provide UV-Vis traces from 300 nm to 700 nm for all spectra. Given substantial gain of knowledge, we rewrote the switching section of the main text:

“After characterizing the photochromic properties of amino DASAs, we investigated the switching capabilities as measured by relative UV-vis absorbance (Figure 4). Absorbance measurements show that amino DASAs **6e** and **7** respond to irradiation with broadband visible light in dichloromethane while only **6e** shows absorbance decrease in toluene (Figure 4a–b). Interestingly, all three generations of amino DASAs have significantly slower conversion rates than their hydroxy counterparts.⁴² In methanol, amino DASAs undergo absorbance decrease but do not recover absorbance upon ceasing irradiation; this is consistent with hydroxy DASA switching behaviors, with notable exceptions.^{43,44} Furthermore, neither **6e** nor **7** display absorbance recovery in any solvent when taken to complete absorbance depletion. When irradiating amino DASAs an absorbance increase in the 300–400 nm range is observed, which correlates with the production of activated pyrrole **4e** ($\lambda_{\text{max}} = 381 \text{ nm}$); this effect is shown in Figure 4c for DASA **6e** in toluene.⁴²

To avoid prolonged irradiation, UV-vis absorbance with *in situ* broadband irradiation across a ten minute interval was measured (Figures 4d and Supplementary Figs. 15–17). Though hydroxy DASAs project switching characteristics respective to their donor and acceptor compositions, most studied amino DASAs fail to switch under these conditions. Only amino DASA **6e** displays productive reversible photoswitching with a 14% forward photoisomerization upon ten minutes of irradiation in toluene and full recovery after discontinuing irradiation (Figure 4d). These results are consistent with faster and reversible photoswitching occurring when the backbone heteroatom is more electron donating, further underscoring the importance of the backbone heteroatom in DASA photoswitching behavior.”

“42. For switching experiments and properties for all other DASAs studied, see Supplementary Fig. 9–17 in the Supplementary Information.”

2. It is interesting that the sulfonamide adducts require such long periods of irradiation to access the closed isomer. Presumably, this is due to the higher energy for the 4π electrocyclization and proton transfer. The authors note that at the onset of the switching experiment, the λ_{max} at 567 nm is higher than the corresponding shoulder at 539 nm,

but after 30 minutes of irradiation the two peaks are observed at equal normalized absorbance. The author should really explore this property in more detail. The broadband white light irradiation might be detrimental to the photoswitching properties of these derivatives. It might be more efficient if narrow band visible light is used, such as 530 nm. Potentially the longer wavelength light is driving the B/B' intermediate back to A and slowing the overall switching efficiency. Monitoring and plotting the kinetics of the 567 and 532 nm independently would also be helpful.

Response: Reviewer #2 brought up great questions and experimental suggestions. As a result, we purchased equipment to perform NMR *in situ* irradiation experiments using a 590 nm narrow band LED, as described by Gschwind [J. Magn. Reson. **232**, 39–44 (2013)]. Unfortunately, the shoulder effect was observed in methanol with amino DASA **6e** and this compound is not sufficiently soluble to perform NMR studies. We tested all three solvents and found that the only solvent amenable to NMR studies was chloroform (λ_{max} in chloroform = 581 nm). No B/B' or switching were observed in chloroform so we were unable to plot the kinetics of the 590 conditions. A control experiment with hydroxy DASA **10** under the same conditions shows complete conversion of **10** to **10'** within 15 minutes.

Though we were unable to track the methanol experiment by NMR, we tracked the absorbance decrease by UV-vis while independently irradiated **6e** in methanol with 490 nm light, 590 nm light, broadband light, and tracked the change with no irradiation. Light source did not influence the relative rate of $\lambda_{\text{max}} = 567$ and $\lambda = 539$ consumption, or pyrrole formation. The experiment with no irradiation also displayed the same level of absorbance decrease and pyrrole formation. Detailed results and experimentation are included in the Supplementary Information (see Supplementary Figs. 9i, 9ii, and 19) and the following was added to the main text:

“In effort to observe triene isomerization and cyclization intermediates, we performed a variable temperature NMR *in situ* irradiation experiment utilizing a narrow band 590 nm LED in chloroform. Narrow light near λ_{max} absorbance rules out potential detrimental effects from broadband exposure. Additionally, amino DASA **6e** displays full absorbance decrease upon irradiation in chlorinated solvents as observed in our UV-vis studies using broadband light (see Supplementary Fig. 9a). Insignificant changes are observed after 72 hours of irradiation with amino DASA **6e** remaining intact (Figure 4g), and no isomers are observed when cooling to -15 °C. Therefore, DASAs bearing a more withdrawing sulfonamide group encounter an unproductive C3–C4 *cis*→*trans* isomerization in chlorinated solvents⁴⁶ at concentrations suitable for NMR acquisition, and prolonged irradiation with broadband light leads to eventual decomposition (Figure 4a and Supplementary Figs. 9–11). Modifications of the backbone heteroatom in combination with additional tuning of other DASA compartments is subject to further investigation.”

“46. Amino DASA **6e** is insufficiently soluble for variable temperature NMR *in situ* irradiation experiments in toluene or methanol.”

3. How efficient is the Z to E isomerization?

Response: We interpreted this question as an extension of question 2. As detailed above, we attempted to follow isomerization of amino DASA **6e** in all three solvents by NMR. Only chloroform provided sufficient resolution due to the insoluble nature of amino DASAs in methanol and toluene. After irradiating amino DASA **6e** in chloroform for 72 hours, we acquired an NMR spectrum of the mixture at -15 °C. Though minor shifts of amino DASA **6e** were observed, no other isomers were detected. Therefore, in chloroform, the Z to E isomerization is not efficient, consistent with the lack of shoulders in chlorinated solvents. Discussion was added to the main text as detailed under question 2 above.

4. It is not clear if the photoswitch recovers after the long reaction times. The authors note that **6e** reaches a photostationary state of 16% after 10 minutes of irradiation and then recovers to 96% open in the dark after 30 minutes. The authors should also report the recovery for DCM, PhMe and MeOH after complete conversion to the closed form (ie. Figure 4b). Does it recover?

Response: This is a great question pertaining to the reversibility of the photoswitches. As such, we performed thermal reversion experiments following irradiation until complete absorbance decrease for each DASA reported. We also measured thermal reversion following a 10 min irradiation period. We found that only amino DASA **6e** in toluene showed any thermal reversion in the 10 min irradiation experiments. None of the amino DASAs showed thermal reversion when irradiation was done until complete absorbance decrease, consistent with degradation. We have

included all experimental details and results in the Supplementary Information (see Supplementary Fig. 9–17). We also added discussion in the main text, as detailed under question 1 above.

5. The authors should show the entire UV-Vis trace from 300 nm to 700 nm for all spectra (at least in the SI). This would prove that the absorbance going down is not due to the formation of the pyrrole 4e. Similar to what is observed in Figure S9.

Response: We thank Reviewer #2 for this suggestion. We have added the entire UV-vis trace to the Supplementary Information as Supplementary Figs. 9–14. We learned of an absorbance increase in the 300–400 nm range for 1) 6e in all solvents, consistent with pyrrole starting material, 2) 7 in dichloromethane and methanol, and 3) 8 in methanol. Discussion was added to the main text, as described under question 1 above.

6. It is not clear what is being shown in Figure 4g. In general, the text and Figures are too small and hard to see. Is the decrease at 40 °C due to the formation of the pyrrole 4e for all solvents or is it due to the formation of closed isomer? Only PhMe is shown in the SI.

Response: We thank Reviewer #2 for pointing this out. We adjusted Figure 4 and generally made all schemes and text larger. We included the full range UV-Vis traces of all DASAs in toluene (see Supplementary Fig. 18) after incubating at 40 °C to assess thermal stabilities. We learned that amino DASA 6e does form pyrrole 4e and added the following discussion to the main text:

“In contrast to Feringa’s non-hydroxy triene,¹⁷ amino DASAs displayed complete absorbance decay after extended periods of irradiation. However, the lack of reversion coupled with an absorbance increase in the 300–400 nm range, the thermal stability of amino DASAs was in question. Upon incubating 2nd generation DASAs in toluene at 40 °C with no irradiation, a significant absorbance decrease is observed. This is not the case for 1st and 3rd generation DASAs. Upon complete loss of absorbance at $\lambda_{\text{max}} = 578$ nm for second generation amino DASA 6e, absorbance at 381 nm corresponding to activated pyrrole 4e increased to 49% conversion (Figure 4e).⁴⁵ In direct contrast, second generation hydroxy DASA 10 displayed absorbance decrease without observed activated furan 4_{fur} formation (Figure 4f).”

“45. For yield calculation from absorbance values, see Supplementary Fig. 18 in the Supplementary Information.”

7. The authors state, “curiously, 6e underwent gradual loss of absorbance (not photoabsorbance) even in the absence of irradiation, and a similar absorbance decrease was observed for 10, indicating a general thermal equilibrium shift for second generation DASAs.” This is not a shift in equilibrium, it is conversion of the DASA adduct to the starting material (pyrrole or activated furan). This should be clarified.

Response: We thank Reviewer #2 for this clarification. We did observe starting material 4e formation for amino DASA 6e, however, no furan starting material was observed by UV-Vis for hydroxy DASA 10. The main text now reflects these observations as detailed under question 6 above.

8. Given that heat promotes the conversion to starting material, the authors need to more carefully study how prevalent this reaction is under their photoirradiation conditions. What temperature is being reached upon irradiation with white light under the reaction conditions? Presumably, higher than 40 °C. If so, is starting material being produced? Is this solvent dependent?

Response: This is a great concern from Reviewer #2 that we had ourselves. Due to the heat emitted from a general light fixture, we opted to use a convection-cooled light as our broadband source and a fiber optic LED for our *in situ* irradiation studies. These light sources do not heat the reaction mixtures. To demonstrate this, we irradiated a solution of toluene over the course of 12 hours and observed a temperature change from 26.2 °C to 26.5 °C without fluctuation. This experiment and the results were added to the Supplementary Information.

9. The NMR for 4a should be expanding in the SI and peak assignments given. It is hard to really see the peaks to clearly see the closed isomer. Is the pyrrole starting observed?

Response: We thank the Reviewer #2 for noting this. Due to ineffective switching of amino DASAs, no switching is observed at the concentrations required for NMR acquisition. The prior NMR study was done by irradiating 5 mg of

amino DASA **6e** in 1 liter of toluene for six days. Aliquots were taken, concentrated, and transferred into chloroform for NMR acquisition.

In efforts to get a better NMR stack, we reperformed this NMR experiment with the newly acquired *in situ* irradiation equipment, which is the field standard. Unfortunately, no isomerization is observed in chloroform and signals are not resolved in methanol or toluene. A full spectrum of the NMR stack in chloroform with peak assignments is provided in the Supplementary Information.

10. In the introduction the authors should expand (briefly) the connection between using a furylcarbinol (aza-Piancatelli and Piancatelli; Read de Alaniz and Piancatelli), furfural (Stenhouse adducts; Stenhouse, Honda, Lewis & Mulquiney) and activated furan (DASA; Safar and Read de Alaniz). This will help place in context the wide-use of furan as the core building block and the unique nature of this work.

Response: This is a great point by Reviewer #2 that strengthens the synthetic aspect of our contribution. The new rearrangement can be used beyond amino DASA synthesis. As such, we included the following descriptions in the main text:

“Overall, the newly discovered pyrrole rearrangement enables the study of the DASA backbone heteroatom compartment and furthers our insight into the structure-property relationship of this complex photoswitch.”

“Pioneering furan rearrangements dating back to as early as 1850 on furfurals by Stenhouse,¹⁸ Honda,¹⁹ and Lewis & Mulquiney,²⁰ furylcarbinols by Piancatelli²¹ and Read de Alaniz,^{22,23} and activated furans by Šafář²⁴ and Read de Alaniz,¹ have contributed to the synthesis of Stenhouse Salts, a multitude of prostanoid acid natural products,²⁵ and DASA molecular photoswitches. Despite their impact, each rearrangement variant has remained limited to furan cores, constraining products to oxygen-containing molecules. Specific to DASAs, only a hydroxy group has been achievable for the backbone heteroatom compartment due to the synthetic dependence on activated furans.”

“Realization of such a transformation would enable the reinvention of an amino equivalent for all hydroxy permutations since 1850, including the more recent DASA syntheses.”

“In conclusion, we report the first pyrrole-based aza-Piancatelli rearrangement, a ¹³C NMR predictive tool for rearrangement viability, and their application to develop a new subclass of donor-acceptor Stenhouse adducts. Confirmed by single crystal X-ray diffraction, amino DASAs enable tunability of the backbone heteroatom, allowing a direct comparison of sulfonamide- and hydroxy-substituted DASAs from each synthetic generation. In comparison to their hydroxy counterpart, amino DASAs produce potential noncovalent interactions in addition to varying electronic contributions that result in a hypsochromic absorbance shift and inefficient photoswitching. It was discovered that substituting the hydroxy group for a sulfonamide moiety results in a decreased molar absorption coefficient and thermal stability, emphasizing the heteroatom’s role in efficiently stabilizing the photoswitch. In combination with previous hydroxy DASA studies, we look forward to the future development of well-behaved amino DASAs. With distinct physical properties and an additional bonding orbital, we expect amino DASAs to open new opportunities for computational studies and applications to better understand these complex photoswitches.”

“25. Verrier, C.; Moebs-Sanchez, S.; Queneau, Y.; Popowycz, F. The Piancatelli Reaction and its Variants: Recent Applications to High Added-Value Chemicals and Biomass Valorization. *Org. Biomol. Chem.* **16**, 676–687 (2018).”

Reviewer #3

Summary:

The paper by Reyes et al. reports the synthesis and study of a new type of donor-acceptor Stenhouse adducts (DASA). DASA photoswitches have received considerable attention due to their interesting properties. However, generalization and use of these photoswitches has been typically hampered by the difficult control of the properties and the challenging synthesis of significantly diverse compounds. This paper presents the synthesis of a different type of DASA. The revised synthetic route allowed the preparation of currently unknown amino analogues of the DASA basic core. The modular building of these molecules has allowed the modification of most parts of the structure, leaving untouched the hydroxy group coming from the furan moiety. In this paper, a clever modification and tuning of the synthesis has allowed the preparation of the new amino derivatives.

The formal aspects of the manuscript are adequate; well-written text, clear and informative figures, and a good selection of references to put in context the research. The methodology used is also correct and the experiments are well-planned and described in detail.

Unfortunately, despite the efforts shown in this paper, the results are very disappointing. Comparison between hydroxy and amino equivalent leads to poor performance of the new compounds for almost every property: hypsochromic absorbance, slower photoconversion and back-reaction. Thus, beyond the modification in the synthetic route allowing for the preparation of new derivatives, results in this paper are mainly negative from the point of view of the application or use of these DASAs.

Overall, the paper is relevant and should be published, but I do not think that the results shown match the requirements to be published in Nature Communications.

Response: We thank Reviewer #3 for their valuable perspective. They correctly described the synthesis of a different type of DASA, which unlocks the final remaining static component of the DASA structure. Also, they pointed out that this synthesis was only possible due to the introduction of the pyrrole rearrangement. We would like to stress that multiple related rearrangement had been limited to furan starting materials since 1850 and the use for DASA synthesis is only one application.

Though Reviewer #3 is correct to state that the amino DASAs characterized have a hypsochromic absorbance and slower switching reactions, 1) desired absorbance and rate of isomerization are typically optimized to be application specific, 2) absorbance and rate of isomerization are tunable with other compartmental/structural changes and having the amino functional group introduces an additional functional handle and tuning element, 3) the purpose of this study was to develop a route and to demonstrate the ability to create and characterize amino DASAs, not to create one with specific absorbance and rate of isomerization features – the *p*-Br sulfonamide was actually selected to facilitate characterization of the open amino DASA through crystallization efforts, and 4) we accomplished our goal and began to better understanding the backbone heteroatom effect, which can take DASAs in new directions, as pointed out by Reviewers #1 and #2.

Overall, we are incredibly thankful for suggestions provided by Reviewer #3 because they are influential for future directions of amino DASA design and studies, including synthetic, mechanistic, and computational, for tunable function.

REVIEWERS' COMMENTS

Reviewer #2 (Remarks to the Author):

The authors have adequately addressed all major concerns, and I support accepting the manuscript pending very minor edits.

1. Consider revising “have contributed to the synthesis of Stenhouse Salts, ...” to “have contributed to the synthesis of Stenhouse salts derivatives, ...” Although a minor change, I believe Stenhouse salts specifically refer to adducts bearing an amine functionalized donor and acceptor groups, with furlycarbinols and activated furans resulting in the formation of unique derivatives.

2. Consider moving the NMR Figure 4g to the SI. While the NMR studies aimed at observing the Z to E is appreciated, given that no results were observed, I believe relocating this part of the Figure to the SI would not detract from the overall manuscript and would avoid possible confusion.

Revisions to NCOMMS-24-12012A, Development and characterization of amino donor-acceptor Stenhouse adducts

Reviewer #2

Summary:

The authors have adequately addressed all major concerns, and I support accepting the manuscript pending very minor edits.

Comments:

1. Consider revising “have contributed to the synthesis of Stenhouse Salts, ...” to “have contributed to the synthesis of Stenhouse salts derivatives, ...” Although a minor change, I believe Stenhouse salts specifically refer to adducts bearing an amine functionalized donor and acceptor groups, with furlycarbinols and activated furans resulting in the formation of unique derivatives.

Response: We agree with Reviewer #2 that this distinction should be made. We made the proper adjustment.

2. Consider moving the NMR Figure 4g to the SI. While the NMR studies aimed at observing the Z to E is appreciated, given that no results were observed, I believe relocating this part of the Figure to the SI would not detract from the overall manuscript and would avoid possible confusion.

Response: We had previously left Figure 4g in the main text because it was an experiment suggested by Reviewer 2. We fully agree and are happy that Reviewer 2 suggests that it should be moved to the SI. This was done.